# Raman Natural Gas Analyzer: Effects of Composition on Measurement Precision

**DOI:** 10.3390/s22093492

**Published:** 2022-05-04

**Authors:** Dmitry V. Petrov, Ivan I. Matrosov, Alexey R. Zaripov, Aleksandr S. Tanichev

**Affiliations:** 1Institute of Monitoring of Climatic and Ecological Systems, 634055 Tomsk, Russia; mii@imces.ru (I.I.M.); alexey-zaripov@rambler.ru (A.R.Z.); tanichev_aleksandr@mail.ru (A.S.T.); 2Department of Optics and Spectroscopy, Tomsk State University, 634050 Tomsk, Russia

**Keywords:** Raman spectroscopy, gas analysis, natural gas, methane, alkanes, isotopic composition, heating value

## Abstract

Raman spectroscopy is a promising method for analyzing natural gas due to its high measurement speed and the potential to monitor all molecular components simultaneously. This paper discusses the features of measurements of samples whose composition varies over a wide range (0.005–100%). Analysis of the concentrations obtained during three weeks of experiments showed that their variation is within the error caused by spectral noise. This result confirms that Raman gas analyzers can operate without frequent calibrations, unlike gas chromatographs. It was found that a variation in the gas composition can change the widths of the spectral lines of methane. As a result, the measurement error of oxygen concentration can reach 200 ppm. It is also shown that neglecting the measurement of pentanes and n-hexane leads to an increase in the calculated concentrations of other alkanes and to errors in the density and heating value of natural gas.

## 1. Introduction

Natural gas (NG) is the most environmentally friendly of all fossil fuels and is also a raw material for the production of many chemicals, including hydrogen [1]. To date, the basic method for measuring its composition is gas chromatography. However, this method has some disadvantages. Among them are the need for consumables, frequent calibration checks, and a long analysis time. These features make real-time measurements impossible. Devices based on optical spectroscopy do not have such drawbacks. The application of infrared (IR) spectroscopy for the analysis of NG composition was demonstrated by Kireev et al. [2,3]. The measurement accuracy of hydrocarbons is close to the gas chromatography. However, it is impossible to measure the content of diatomic homonuclear molecules (such as N_2_, O_2_, H_2_, etc.), using this method. Taking into account the ongoing development of energy technologies with minimal CO_2_ emissions, the use of hydrogen-enriched natural gas will increase [4,5]. In this regard, IR spectroscopy is not an ideal method for measuring such gas mixtures. Raman spectroscopy is a promising alternative technique. It is possible to simultaneously control the content of all types of molecules using an instrument based on this effect. The capabilities of such gas analyzers were demonstrated in many studies [6,7,8,9,10,11,12,13,14,15,16,17]. It should be noted that many authors measure alkanes only up to C4. This is explained by the weakness of the Raman signals of gaseous components and the difficulty in deriving the concentrations of heavy alkanes from the Raman spectrum of NG due to the significant overlap of the spectra of various components [18]. According to ISO 6974-5 [19], the detection limit for C2–C6 alkanes is 0.005%. Thus, a Raman gas analyzer must measure NG composition with this accuracy to be competitive with gas chromatographs. In this work, we study the capabilities of the developed Raman gas analyzer using NG samples whose composition varies in ranges close to values indicated in ISO 6974-5 [19]. In addition, we investigate the influence of line broadening and the effect of ignoring the spectra of C5+ alkanes on measurement precision.

## 2. Materials and Methods

### 2.1. Raman Gas Analyzer

The Raman gas analyzer used in this work is an improved analog of the device as that used previously [6]. Its optical design is based on a 90-degree geometry of scattered light collection (see Figure 1) since spectra with a minimum background level can be recorded using this scheme. A solid-state continuous-wave laser with a power of 1.5 W at a wavelength of 532 nm was used as a source of exciting radiation. Two identical f/1.8-lenses were used for scattered light collection. An analysis of our previous results [6] and the Raman spectra of the main NG components [18] showed that it is necessary to improve the signal-to-noise ratio to improve the accuracy of measurements. In this regard, a new compact no-moving-parts f/1.8-spectrometer MKR-2m (Sibanalitpribor LLC, Tomsk, Russia) was used in this work. Its main difference from the previous spectrometer [6] is a higher spectral sensitivity (especially at the edges of the recorded range) due to the optimization of the optical scheme. The simultaneously recorded spectral range was 530–628 nm using the 1800 lines/mm grating. With an entrance slit of 40 µm, the half-width of instrumental function response was ~6 cm^−1^ at the center of this range. The signals were recorded using the charge-coupled device (CCD) sensor Hamamatsu S10141 (2048 × 256 pixels, 12 µm in size) with thermoelectric cooling down to −10°C. About 10-fold amplification of the Raman signals was obtained in the range of 300–1000 cm^−1^, where the characteristic peaks of C2+ alkanes are located, using this spectrometer (in comparison with Ref. [6]).

### 2.2. Concentration Measurement Method

The contour fit method was used to derive the concentrations due to the significant overlap in the spectra of NG species [18]. Its essence is as follows. The NG spectrum *I_mix_*(ν) at each wavenumber ν can be represented as the sum of the spectra of its components *I_i_*(ν):(1)$$I_{mix}(\nu )=\sum_{i=1}^{m}a_{i}I_{i}^{}(\nu ),$$
where *a_i_* is the contribution of the spectrum of the *i*th component to the spectrum of the mixture [0..1], and *m* is the number of measured components.

Taking into account the number of CCD sensor columns, a system of 2048 equations can be obtained. Its solution (contributions *a_i_*) can be found using the least-squares method. The required relative concentrations (*N_i_*) can be found using Equation (2).
(2)$$N_{i}=\frac{n_{i}a_{i}}{\sum_{j=1}^{m}n_{j}a_{j}}\cdot 100\%,$$
where *n_i_* is the absolute concentration of the *i*th component in the reference spectrum *I_i_*(ν).

According to Ref. [20], the spectral characteristics (peak positions and half-widths) of the reference spectra and the spectra of the mixture should be equivalent to obtain the most accurate results. First of all, to ensure this condition, all measurements of mixtures were carried out at a pressure of 25 atm and a temperature of 300 K. Reference spectra of pure methane, ethane, nitrogen, carbon dioxide, hydrogen, and oxygen were also obtained at these parameters. The spectra of heavier alkanes (propane, n-butane, isobutane, n-pentane, iso-pentane, neo-pentane, and n-hexane) liquefy under the above conditions. For this reason, they were obtained at saturated vapor pressure. The exposure time for each reference spectrum was 1000 s.

### 2.3. Experiment

Three samples of synthetic NG with significantly different compositions were used for research (see Table 1). These samples are the reference gas mixtures with low uncertainties that were purchased from Monitoring LLC (Saint Petersburg, Russia). Measurements were carried out for three weeks, once a week, to assess the long-term stability of the results. The sequence of analysis of mixtures is presented in Table 2. A series of five measurements were performed for each mixture with the replacement of the sample in the cell. The time of one analysis was 30 s. Note that the set of reference spectra of pure components was obtained once before the measurement procedure was started. Additional calibration procedures were not performed during all measurements.

## 3. Results and Discussion

### 3.1. Mixture Measurements

Figure 2 and Figure 3 show the obtained Raman spectra of the samples of NG. Despite mutual overlaps, the characteristic peaks of most components are distinguishable at the resolution of the spectrometer used. The achieved sensitivity makes it possible to see the lines of the ν_4_ band of methane down to ~800 cm^−1^. In addition, a wide unresolved band is observed in the methane spectrum in the region of 300–600 cm^−1^. We suppose this is a collision-induced rotational band [21,22], which is attenuated up to ~350 cm^−1^ by the notch filter. Bands of C–C–C deformation vibrations of C3+ hydrocarbons are also located in the region of 300–500 cm^−1^ (see Figure 4). The accuracy of concentration measurements can be improved using this range due to intense peaks of n-butane (429 cm^−1^), n-pentane (398 cm^−1^), and iso-pentane (459 cm^−1^), the overlap of which is not as significant as in the region of 700–1000 cm^−1^. Thus, to measure low concentrations, it is necessary to take into account the contribution of the methane spectrum to the spectrum of NG not only in the region of >990 cm^−1^ (as indicated in Ref. [18]) but also in the region of lower wavenumbers. The inset in Figure 2 shows the vibrational band of nitrogen (2330 cm^−1^), whose concentration in sample 1 is 54 ppm, despite its significant overlap with the lines of the 2ν_4_ and ν_3_ bands of methane, is also well observed. Hence, concentrations with a sensitivity of <50 ppm can be measured due to the achieved signal-to-noise ratio. The limits of detection will be estimated below.

The range of 300–2400 cm^−1^ was used to determine the composition of mixtures. All measured concentrations during one day for each mixture were averaged. The concentrations (C) and their standard deviations (σ) are presented in Table 3, Table 4 and Table 5. It can be seen that the measured and reference concentrations are in good agreement taking into account the uncertainties. The only exception is data of n-hexane in samples 2 and 3. For most components, the variation in measured concentrations over all days is within their mean standard deviation. It indicates these variations are due to noise in the spectra. Thus, the presented data confirm that Raman gas analyzers can operate for a long time without calibration, unlike gas chromatographs.

The relative measurement errors of each component were obtained using the mean standard deviations (see Figure 5). It can be seen that these values depend both on the concentration and the type of molecule (due to different scattering cross-sections and the level of overlap of the spectral bands). Taking into account that the measurement errors of gas chromatographs are close to 5%, it can be concluded that the accuracy of the presented Raman gas analyzer is higher for species with a concentration of more than ~100 ppm.

### 3.2. Limits of Detection

Limits of detection (*LOD_i_*) were estimated using Equation (3). Here, we defined the concentrations at which the signal of *i*th component is three times the standard deviation of the noise. The spectrum of sample 1 was used to obtain these data. Peak intensities of each component (*S_i_*) were estimated, taking into account their contribution to the spectrum of the mixture (see Figure 6). The difference between two successive spectra of sample 1 was obtained to estimate the magnitude of the noise (see Figure 7). It can be seen that the noise in the region of 500–1000 cm^−1^, where the characteristic bands of C2+ alkanes are located, is less than in the region of intense lines of the ν_2_ band of methane (1200–1700 cm^−1^). This feature is related to the effect of photon shot noise, which is proportional to the square root of the signal intensity. In this regard, the noises that affect measurement errors and *LOD*s are higher for CO_2_ and O_2_ than for all other components. The standard deviations of noise (*N_i_*) were calculated using the intensities in the spectrum shown in Figure 7 in the following regions: 1540–1580 cm^−1^ (for O_2_), 1280–1380 cm^−1^ (for CO_2_), and 700–900 cm^−1^ (for other components). Concentrations of components (*C_i_*) in sample 1 for calculations were taken from Table 1. The results obtained are presented in Table 6. It can be seen that the *LOD* values are within the range of 2–35 ppm. Thus, the achieved sensitivity of the Raman analyzer meets the requirements of ISO 6974-5 [19].
(3)$$LOD_{i}=3\frac{C_{i}}{S_{i}/N_{i}},$$

### 3.3. Influence of Line Broadening on Measurements

Let us consider the features of O_2_ measurement. It has one fundamental vibrational band with the position of the maximum at 1555 cm^−1^, which is overlapped by the ν_2_ band of methane (see Figure 3). Hence, the measurement accuracy is affected by the broadening of the spectral lines of methane [20] besides the signal-to-noise ratio. Pressure [23] and molecular environment [24,25] influence the half-widths of the lines. The line at 1793 cm^−1^ was analyzed to assess the influence of the composition on the line half-widths of the ν_2_ band of methane. This line was chosen since it is not overlapped by the spectra of other species and, therefore, the measurement error of its half-width in mixtures is eliminated. The data obtained and the half-width of this line as a function of pure methane pressure are shown in Figure 8. It can be seen that the half-width increases with a decrease in the fraction of methane in the mixtures. This broadening is related to an increase in the concentration of heavy hydrocarbons in the mixture since the methane-methane broadening coefficients are less than the broadening coefficients of methane-ethane, methane-propane, etc. [25].

According to Figure 8, an increase in the pressure of pure methane to 26.6 atm leads to the same broadening as in the spectrum of sample 3 at a pressure of 25 atm. Thus, in our case, we can use the spectra of pure methane at pressures of 25.0 and 26.6 atm to estimate the error in oxygen measurements due to the broadening of methane lines. The spectrum at a pressure of 26.6 atm was multiplied by the 25/26.6 value to ensure equal integral intensities of these spectra. Figure 9 shows the difference between these methane spectra in the region of 1555 cm^−1^, denoted as *R*. According to Equation (4), this effect leads to an oxygen measurement error (∆) close to 200 ppm.
(4)$$\Delta =\frac{R\cdot 100\%}{I_{MAX}},$$
where *I_MAX_* is the peak intensity of the spectrum of pure oxygen at 25 atm. Taking into account the concentration ranges of C2+ alkanes in NG [19], it can be concluded that the systematic error in oxygen measurement can reach 200 ppm (depending on the composition). This error is less than the uncertainty of the reference O_2_ concentration in sample 3. However, in the case of an O_2_ concentration in such a mixture below 200 ppm, this is a sufficiently large value that cannot be ignored. Calibration coefficients or a reference spectrum of pure methane at a pressure that results in the required line broadening can be used to obtain reliable data.

We believe that the deviations of the measured hexane concentrations from the reference values are due to similar effects. Although hexane has several bands in the region of 700–900 cm^−1^, their peak intensity is relatively low (see Figure 6), and all of them are overlapped by the spectra of other molecules [18]. Thus, a change in the spectral characteristics of alkanes in a mixture compared to a pure substance can lead to errors in measurements of the hexane concentration. We plan to study these features in more detail in the future.

### 3.4. Estimation of Errors in the Case of Ignoring C5+ Spectra

We decided to estimate the errors in the case of neglecting pentanes and hexane since many authors analyze the composition of mixtures only up to C4 [7,9,10,11,12,13,14,15]. All spectra of mixtures obtained during the first day of experiments were used. The spectra of pentanes and hexane were excluded from the set of reference spectra of pure components to calculate the concentrations. The results obtained are presented in Table 7. It can be seen that ignoring these components leads to an increase in the measured concentrations of ethane, propane, and butanes. Taking into account that this effect is due to the overlap of their spectra, the errors depend on the composition of the mixture and cannot be eliminated using calibration coefficients. In addition to these data, the characteristic parameters [26], which are required for power plant operators, were calculated. To this end, the concentrations shown in Table 7 and Table 3, Table 4 and Table 5 (1st day) were used. As shown in Table 8, these characteristics correspond to the reference data when all components are measured. In turn, only the heating value of sample 1 corresponds to the reference value in the case of ignoring the measurement of pentanes and hexane. Despite the increase in the measured concentrations of other alkanes, other characteristics are significantly less than the reference ones. Thus, reliable characteristic parameters of NG cannot be obtained by measuring alkanes only up to C4.

### 3.5. Variation in the Isotopic Composition of Methane

We noticed the different intensity of the peak with a wavenumber of 2196 cm^−1^ between the spectra of pure methane and sample 1 during the experiments. The ν_2_ band of the CH_3_D methane isotopologue is located in this region (see Figure 10). This discrepancy may be due to the different nature of the origin of the pure methane and methane in the mixtures. The difference in the peak intensity is ~0.4% and agrees with possible CH_3_D/CH_4_ variations in NG [27]. We did not find signs of ^13^CH_4_/^12^CH_4_ variation in our samples since there is a small shift in their lines relative to each other in the ν_2_ region [28]. It is worth noting that knowledge of the isotopic composition of methane is also useful. It is possible to determine the type of reservoir (gas, gas condensate, or oil), as well as the origin of natural gas (biogenic or thermogenic) based on this information [29]. Raman gas analyzer can also measure the content of ^13^CH_4_ by registration of spectra up to 3100 cm^−1^ [30,31]. Note that when using the contour fit method, the discrepancy in the isotopic composition of methane in comparison to the reference methane can lead to a difference between their spectra and, consequently, to errors in the measurement of other components. In this case, the simulation of spectra can be used to improve the reliability of measurements [32]. The effects of pressure, molecular environment, and the contributions of all isotopologues can be taken into account to obtain a spectrum using this approach.

## 4. Conclusions

This study presents the features of natural gas analysis using Raman spectroscopy. The use of the contour fit method to derive concentrations from the spectra of mixtures makes it possible to obtain reliable results even with a significant change in the composition of the samples. However, in the case of measuring low concentrations of components whose characteristic peaks are overlapped by intense bands of other molecules, it is necessary to take into account the change in spectral characteristics due to changes in the molecular environment to increase the accuracy. The data obtained confirmed that such devices can operate for a long time without calibration. This is a very important advantage of Raman gas analyzers over analogs. The achieved detection limits of the developed compact Raman gas analyzer are 2–35 ppm at a pressure of 25 atm and an analysis time of 30 s. This level of sensitivity makes it possible to monitor the isotopic composition of methane. In turn, it is possible to reduce the analysis time or improve the accuracy by using a more powerful laser and/or a photodetector with a lower noise level. Taking into account the advantages of Raman gas analyzers, we believe that they have great potential in natural gas analysis and can replace conventional gas chromatographs.

## Figures and Tables

**Figure 1 sensors-22-03492-f001:**
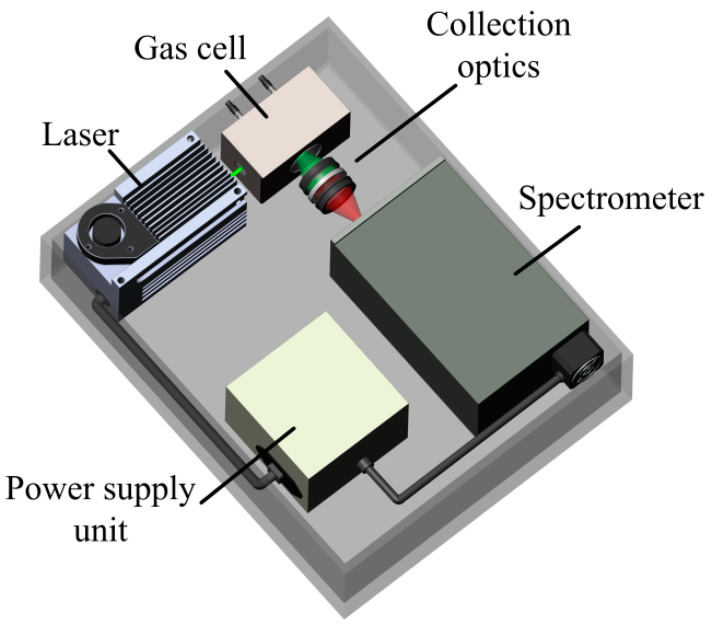
Schematic of Raman gas analyzer.

**Figure 2 sensors-22-03492-f002:**
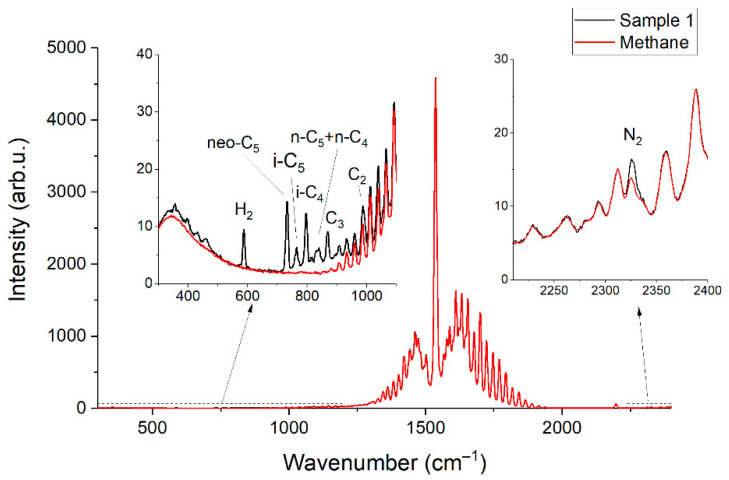
Raman spectra of pure methane and sample 1.

**Figure 3 sensors-22-03492-f003:**
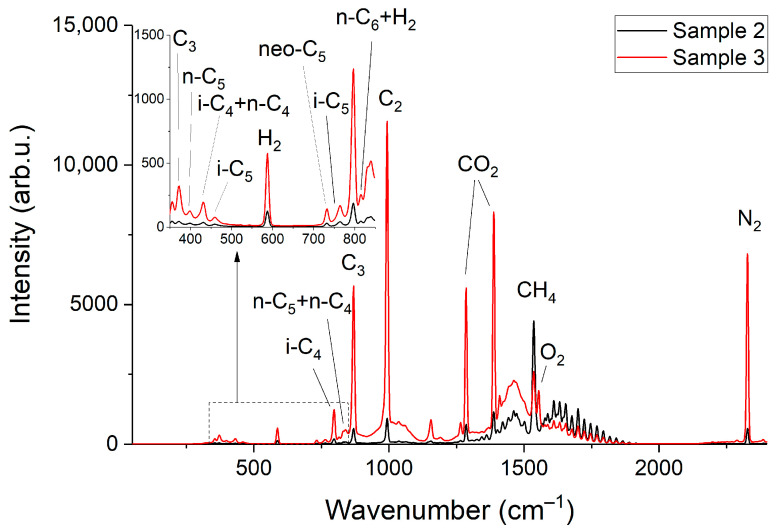
Raman spectra of sample 2 and sample 3.

**Figure 4 sensors-22-03492-f004:**
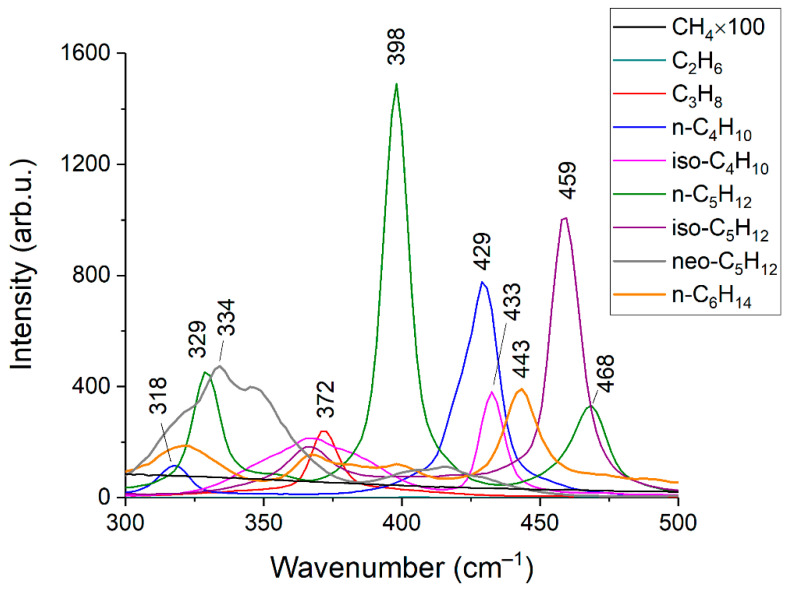
Raman spectra of C1–C6 alkanes in the range of 300–500 cm^−1^. The intensities correspond to the equivalent pressure.

**Figure 5 sensors-22-03492-f005:**
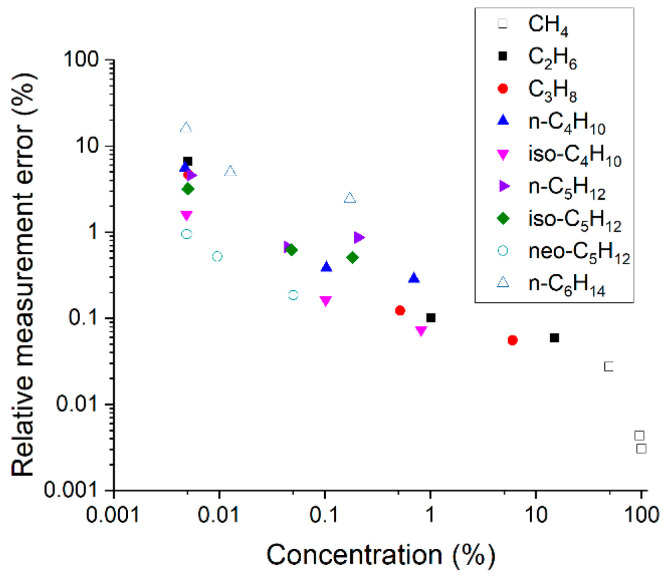
Relative measurement errors of alkanes at different concentrations.

**Figure 6 sensors-22-03492-f006:**
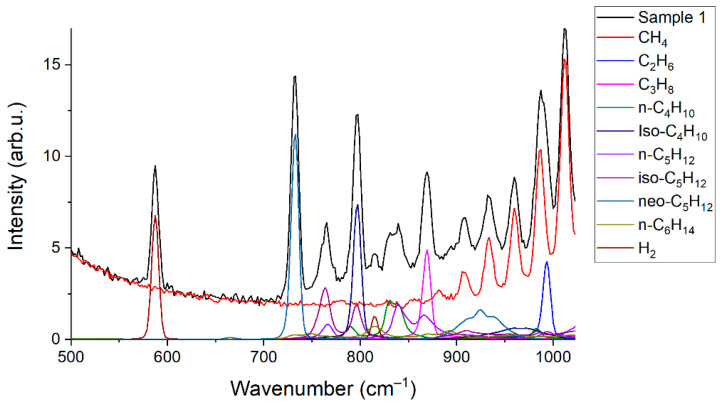
Contributions of the species to the spectrum of sample 1.

**Figure 7 sensors-22-03492-f007:**
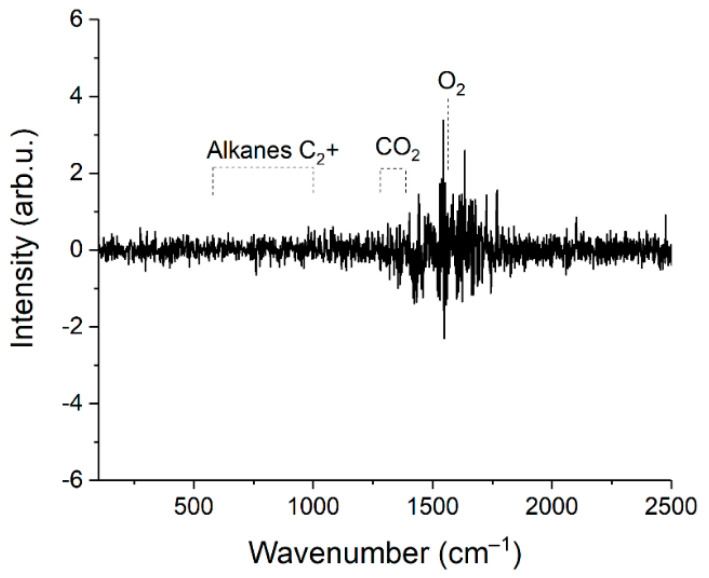
Difference between two successive spectra of sample 1.

**Figure 8 sensors-22-03492-f008:**
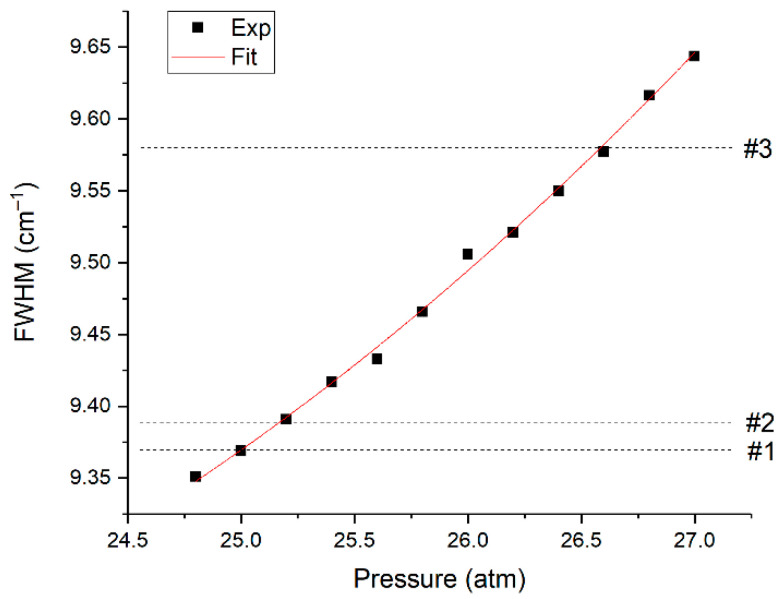
Half-width of the methane line at 1793 cm^−1^ in pure methane at various pressures and in analyzed samples at 25 atm.

**Figure 9 sensors-22-03492-f009:**
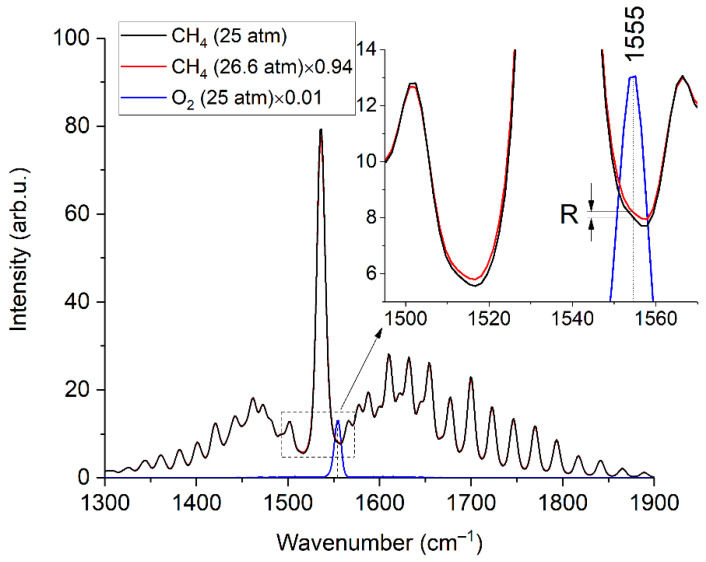
Raman spectra of methane (at 25 and 26.6 atm) and oxygen (at 25 atm). The inset shows that the broadening of the methane lines leads to different intensities in the region where the oxygen band is located.

**Figure 10 sensors-22-03492-f010:**
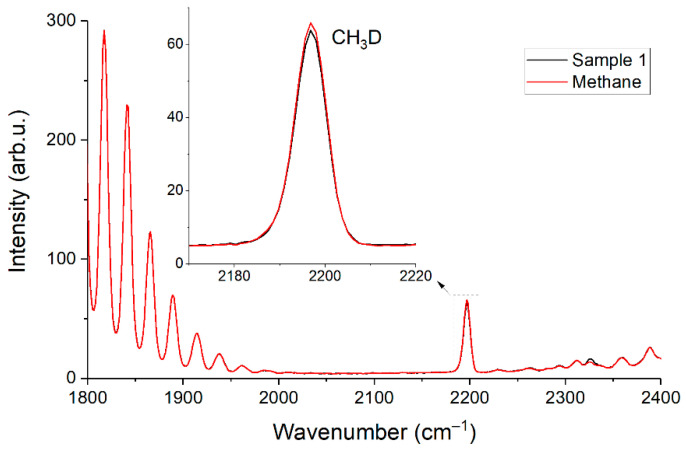
Raman spectra of pure methane and sample 1 in the range of 1800–2400 cm^−1^.

**Table 1 sensors-22-03492-t001:** Composition of natural gas samples used.

Component	Concentration (%)		
	Sample 1	Sample 2	Sample 3
CH_4_	99.9403	95.998	49.0379
C_2_H_6_	0.00496	0.997	15.1
C_3_H_8_	0.00474	0.509	6.05
n-C_4_H_10_	0.00493	0.105	0.709
iso-C_4_H_10_	0.00497	0.102	0.816
n-C_5_H_12_	0.00503	0.0474	0.205
iso-C_5_H_12_	0.00522	0.0472	0.19
neo-C_5_H_12_	0.0048	0.01	0.0511
n-C_6_H_14_	0.00445	0.0236	0.131
CO_2_	0.0047	1	10.1
N_2_	0.0054	1.039	15.1
H_2_	0.00559	0.102	0.5
O_2_	0.0048	0.0198	2.01

**Table 2 sensors-22-03492-t002:** Program of measurements.

Day	Sequence of Sample Analysis
1st	#1–#2–#3–#2–#1–#3–#2
2nd	#2–#1–#2–#1–#2–#1–#2–#3
3rd	#1–#3–#1–#3–#1–#3–#1–#2

**Table 3 sensors-22-03492-t003:** Measurement results for sample 1.

Component	Reference Data		Data Obtained					
			1st Day		2nd Day		3rd Day	
	C (%)	σ (%)	C (%)	σ (%)	C (%)	σ (%)	C (%)	σ (%)
CH_4_	99.9403	0.0023	99.94	0.0023	99.938	0.0022	99.9401	0.0046
C_2_H_6_	0.00496	0.00018	0.00479	0.00027	0.00508	0.00025	0.00526	0.00047
C_3_H_8_	0.00474	0.00022	0.00496	0.00011	0.0052	0.00024	0.0052	0.00031
n-C_4_H_10_	0.00493	0.00023	0.00453	0.00025	0.00501	0.00027	0.00466	0.00030
iso-C_4_H_10_	0.00497	0.00023	0.00492	0.00006	0.0049	0.00007	0.00486	0.00011
n-C_5_H_12_	0.00503	0.00023	0.00545	0.00019	0.00549	0.00018	0.00514	0.00032
iso-C_5_H_12_	0.00522	0.00024	0.00496	0.00019	0.00517	0.00015	0.00508	0.00016
neo-C_5_H_12_	0.0048	0.00023	0.00492	0.00004	0.00493	0.00005	0.00494	0.00005
n-C_6_H_14_	0.00445	0.00021	0.00505	0.00064	0.00524	0.00072	0.00429	0.00098
CO_2_	0.0047	0.0005	0.00527	0.00071	0.00509	0.00031	0.00504	0.0011
N_2_	0.0054	0.0005	0.00539	0.00035	0.0048	0.00027	0.00584	0.0006
O_2_	0.0048	0.0005	0.00457	0.0010	0.00595	0.0011	0.00428	0.0014
H_2_	0.00559	0.00025	0.0051	0.00008	0.00508	0.00008	0.0052	0.00009

**Table 4 sensors-22-03492-t004:** Measurement results for sample 2.

Component	Reference Data		Data Obtained					
			1st Day		2nd Day		3rd Day	
	C (%)	σ (%)	C (%)	σ (%)	C (%)	σ (%)	C (%)	σ (%)
CH_4_	95.998	0.09	95.9512	0.0042	95.9509	0.0046	95.9503	0.0029
C_2_H_6_	0.997	0.02	1.0172	0.0010	1.0181	0.0011	1.0179	0.0009
C_3_H_8_	0.509	0.015	0.5166	0.0006	0.5168	0.0008	0.5173	0.0005
n-C_4_H_10_	0.105	0.003	0.1038	0.0004	0.1035	0.0005	0.1042	0.0003
iso-C_4_H_10_	0.102	0.003	0.1018	0.0002	0.1018	0.0002	0.1019	0.0002
n-C_5_H_12_	0.0474	0.0015	0.0455	0.0003	0.0446	0.0003	0.0446	0.0003
iso-C_5_H_12_	0.0472	0.0015	0.0479	0.0002	0.0481	0.0004	0.0482	0.0003
neo-C_5_H_12_	0.01	0.0004	0.0096	0.00005	0.0096	0.00006	0.0096	0.00004
n-C_6_H_14_	0.0236	0.0008	0.0184	0.0007	0.0183	0.0006	0.0186	0.0006
CO_2_	1	0.03	1.0238	0.0012	1.0234	0.0010	1.0228	0.0006
N_2_	1.039	0.021	1.0447	0.0014	1.0451	0.0015	1.0432	0.0005
O_2_	0.0198	0.001	0.0206	0.0015	0.0205	0.0017	0.0221	0.0008
H_2_	0.102	0.003	0.0989	0.0002	0.0988	0.0002	0.099	0.0001

**Table 5 sensors-22-03492-t005:** Measurement results for sample 3.

Component	Reference Data		Data Obtained					
			1st Day		2nd Day		3rd Day	
	C (%)	σ (%)	C (%)	σ (%)	C (%)	σ (%)	C (%)	σ (%)
CH_4_	49.038	1.12	49.499	0.0285	49.517	0.0049	49.518	0.0071
C_2_H_6_	15.1	0.3	14.908	0.0079	14.913	0.0081	14.905	0.0103
C_3_H_8_	6.05	0.18	6.0128	0.0036	6.0138	0.0021	6.0091	0.0043
n-C_4_H_10_	0.709	0.021	0.6987	0.0024	0.6985	0.0019	0.698	0.0017
iso-C_4_H_10_	0.816	0.025	0.8177	0.0005	0.8175	0.0006	0.817	0.0007
n-C_5_H_12_	0.205	0.006	0.204	0.0015	0.209	0.0017	0.2089	0.0022
iso-C_5_H_12_	0.19	0.006	0.1832	0.001	0.1829	0.0009	0.1828	0.0009
neo-C_5_H_12_	0.0511	0.0016	0.0502	0.0001	0.0502	0.0001	0.0508	0.0001
n-C_6_H_14_	0.131	0.004	0.1444	0.0033	0.1564	0.0044	0.1566	0.0049
CO_2_	10.1	0.3	9.9551	0.0151	9.931	0.0106	9.9319	0.0102
N_2_	15.1	0.3	15.035	0.015	15.02	0.0078	15.032	0.0134
O_2_	2.01	0.06	1.978	0.0017	1.9772	0.0007	1.9766	0.0012
H_2_	0.5	0.015	0.5141	0.0008	0.5134	0.0006	0.5125	0.001

**Table 6 sensors-22-03492-t006:** Parameters for Equation (3) and limits of detection of the Raman natural gas analyzer.

Component	*S* (arb.u.)	*N* (arb.u.)	*LOD* (ppm)
C_2_H_6_	4.3	0.017	5.9
C_3_H_8_	4.94	0.017	4.9
n-C_4_H_10_	2.15	0.017	11.7
iso-C_4_H_10_	7.39	0.017	3.4
n-C_5_H_12_	1.94	0.017	13.2
iso-C_5_H_12_	2.87	0.017	9.3
neo-C_5_H_12_	11.27	0.017	2.1
n-C_6_H_14_	0.73	0.017	31.1
CO_2_	5.5	0.036	9.2
N_2_	2.8	0.017	9.8
O_2_	2.8	0.068	35.1
H_2_	6.8	0.017	4.2

**Table 7 sensors-22-03492-t007:** The results of the analysis of mixtures, the spectra of which were obtained during the first day of experiments, in the case of ignoring C5+ alkanes. C*/C is the ratio of the concentration obtained by measuring alkanes up to C4 to the concentration obtained by measuring all components (data from Table 3, Table 4 and Table 5).

Component	Sample 1		Sample 2		Sample 3	
	C* (%)	C*/C	C* (%)	C*/C	C* (%)	C*/C
CH_4_	99.9419	1.000	95.960	1.000	49.779	1.006
C_2_H_6_	0.00644	1.353	1.0280	1.011	14.872	0.998
C_3_H_8_	0.0088	1.774	0.5405	1.046	6.1022	1.015
n-C_4_H_10_	0.01348	2.975	0.1588	1.530	0.9894	1.416
iso-C_4_H_10_	0.00746	1.516	0.1194	1.173	0.8907	1.089
n-C_5_H_12_	--	--	--	--	--	--
iso-C_5_H_12_	--	--	--	--	--	--
neo-C_5_H_12_	--	--	--	--	--	--
n-C_6_H_14_	--	--	--	--	--	--
CO_2_	0.00629	1.194	1.0273	1.003	9.9140	0.996
N_2_	0.0055	1.020	1.0443	0.999	14.963	0.995
O_2_	0.00447	0.978	0.0199	0.966	1.9616	0.992
H_2_	0.00561	1.100	0.1016	1.027	0.5285	1.028

**Table 8 sensors-22-03492-t008:** Comparison of characteristics of natural gas samples.

Sample	Parameter	Reference Data	Data Obtained	
			All Species Were Measured	C5 and C6 Were Ignored
1	Lower heating value (MJ/kg)	33.45 ± 0.03	33.45	33.44
	Relative density	0.55545 ± 0.00004	0.55547	0.55528
2	Lower heating value (MJ/kg)	33.54 ± 0.04	33.53	33.47
	Relative density	0.5838 ± 0.0004	0.5841	0.5830
3	Lower heating value (MJ/kg)	33.12 ± 0.19	33.13	32.86
	Relative density	0.8908 ± 0.0040	0.8876	0.8807

## Data Availability

Data are contained within the article.

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
