# Peer review of "Raman Natural Gas Analyzer: Effects of Composition on Measurement Precision"

_sensors, 2022, doi:10.3390/s22093492_

Round 1
Reviewer 1 Report
In my previous work, I have carried out experimental sample component tests using Raman spectroscopy technology sometimes, but I never concern about details of precision check. This manuscript has done a meaningful work in the determination of NG composition compared with gas chromatographs. Before it is accepted for publication, there still have several uncertainties which need to be further clarified.
The main comments are as follows:
1. The contour fit method, used for deriving the concentrations due to the significant overlap in the spectra of NG species, is not introduced. It is better to give a brief description of the method, although which is detailed in Ref.6.
2. Have you ever determined the content of natural gas in the samples using other method, for example, GC or GC-MS? The concentration of different gas in samples is vital, but it may be different from the theory value.
3. As it is known to us all, the intensity of Raman peak is influenced by varies kinds of factors, such as the stability of laser. Taking account of the factors of Raman test, the difference in peak intensity is too low to tell any difference between CH3D and CH4 (fig.10).
4. Maybe the samples in this ms cannot reflect the natural gas composition in industry or in natural field. For example, we all know that the natural gas samples in oil and gas production are likely to contain sulfur component. Will these situations affect your conclusion?
5. The comparison in this ms mainly involves different gas components, and several pressure scopes are also compared. How accurate would it be if the pressure range has a bit wider? Can you compare samples at different temperatures?
6. There are several grammar mistakes in the manuscript. And, I am not personally accustomed to certain ways of writing on references in the text, such as lines 49-50. In my opinion, maybe expressions on authors or concrete research finding/conclusion of references are better for readers to know what your wish to state in ms.
Author Response
The main comments are as follows:
- The contour fit method, used for deriving the concentrations due to the significant overlap in the spectra of NG species, is not introduced. It is better to give a brief description of the method, although which is detailed in Ref.6.
- The description has been added.
- Have you ever determined the content of natural gas in the samples using other method, for example, GC or GC-MS? The concentration of different gas in samples is vital, but it may be different from the theory value.
- In this work, reference gas mixtures with low uncertainty were used. Such gas mixtures are used to calibrate gas chromatographs or other gas analyzers. In this regard, the measurement of their composition using a GC or GC-MS is incorrect.
- As it is known to us all, the intensity of Raman peak is influenced by varies kinds of factors, such as the stability of laser. Taking account of the factors of Raman test, the difference in peak intensity is too low to tell any difference between CH3D and CH4 (fig.10).
- We do not agree. When the laser power is varied, the intensities of all lines in the Raman spectra change simultaneously. The spectra in Fig. 10 have equal intensity in the region of v2 lines of methane (1535 cm-1). In this regard, the ratio of intensities and, consequently, the concentrations of CH3D/CH4 in the presented spectra are different.
- Maybe the samples in this ms cannot reflect the natural gas composition in industry or in natural field. For example, we all know that the natural gas samples in oil and gas production are likely to contain sulfur component. Will these situations affect your conclusion?
- Taking into account that the most intense peaks of sulfur components (hydrogen sulfide and thiols) are located in the region of 2600 cm-1, their appearance in the analyzed sample will not have a significant effect on the intensities of the components that were measured in this work. In turn, to measure such species using the Raman gas analyzer, it is sufficient to add the spectra of pure sulfur components to the set of reference spectra.
- The comparison in this ms mainly involves different gas components, and several pressure scopes are also compared. How accurate would it be if the pressure range has a bit wider? Can you compare samples at different temperatures?
- In this manuscript, we estimated only the measurement error of oxygen. According to the data presented a pressure change of 1.6 atm (without significant changes in composition) leads to an oxygen measurement error of 200 ppm. Therefore, a pressure change of x atm will approximately lead to a measurement error of x*200/1.6 ppm. At this moment, we are unable to carry out measurements at different temperatures due to the lack of a heated gas cell. In turn, a temperature change will also lead to an increase in the measurement error due to changes in the line intensities.
- There are several grammar mistakes in the manuscript. And, I am not personally accustomed to certain ways of writing on references in the text, such as lines 49-50. In my opinion, maybe expressions on authors or concrete research finding/conclusion of references are better for readers to know what your wish to state in ms.
- The text and writing of references have been improved.
Reviewer 2 Report
- Raman spectroscopy was used to measure the components of natural gas samples with three various CH4 The operating characteristics of measurement by Raman spectroscopy were compared with gas chromatographs. While the experimental measurement results could provide helpful reference for relevant academicians, a few modifications are suggested prior to further being considered for possible publication in the journal.
- Whole words should appear when their corresponding abbreviations first appear, for example: IR at line 29. In addition, FWHM is not an acronym for “half-width of instrumental function response”.
- In Section 2: Materials and methods, please supplement the method to determine Limits of detection (LOD), which results are presented in Table 6. More detailed discussion for LOD in Table 6 is suggested in Section 3.2 on page 7.
- Please provide more details for “Monitoring LLC” at line 82.
- In Table 8, Lower heating value is suggested to replace present “Inferior calorific value” on page 11. The unit of lower heating value such as (MJ/kg) could be provided as well. In addition, what are the compositions of Mixture 1 ~ 3? Are they the same as Sample 1 ~ 3 in Table 7?
Author Response
- Raman spectroscopy was used to measure the components of natural gas samples with three various CH4 The operating characteristics of measurement by Raman spectroscopy were compared with gas chromatographs. While the experimental measurement results could provide helpful reference for relevant academicians, a few modifications are suggested prior to further being considered for possible publication in the journal.
- Whole words should appear when their corresponding abbreviations first appear, for example: IR at line 29. In addition, FWHM is not an acronym for “half-width of instrumental function response”.
- We added the description before the acronym «IR» and we deleted acronym «FWHM».
- In Section 2: Materials and methods, please supplement the method to determine Limits of detection (LOD), which results are presented in Table 6. More detailed discussion for LOD in Table 6 is suggested in Section 3.2 on page 7.
- We added more information to the Section 3.2 and Table 6.
- Please provide more details for “Monitoring LLC (Saint Petersburg, Russia)” at line 82.
- Details have been added. “Monitoring LLC (Saint Petersburg, Russia)”
- In Table 8, Lower heating value is suggested to replace present “Inferior calorific value” on page 11. The unit of lower heating value such as (MJ/kg) could be provided as well. In addition, what are the compositions of Mixture 1 ~ 3? Are they the same as Sample 1 ~ 3 in Table 7?
- We replaced these terms. Yes, the compositions are the same. The terms "Mixture 1 ~ 3" have been replaced by the terms "Sample 1 ~ 3" in the table 7 and in the text.